Bayesian estimation for the mean of delta-gamma distributions with application to rainfall data in Thailand

Kaewprasert Theerapong
http://orcid.org/0000-0001-8269-3397 Niwitpong Sa-Aat
http://orcid.org/0000-0003-3059-1131 Niwitpong Suparat suparat.n@sci.kmutnb.ac.th
Department of Applied Statistics, Faculty of Applied Science, King Mongkut’s University of Technology North Bangkok , Bangkok , Thailand
Wang Lei
Electronic publication date: 2022 May 18
Publication date: 2022
Volume: 10
Electronic Location ID: e13465
Received 2022 Feb 16; Accepted 2022 Apr 28
Copyright: © 2022 Kaewprasert et al.
Copyright year: 2022
Copyright holder: Kaewprasert et al.
License: This is an open access article distributed under the terms of the Creative Commons Attribution License, which permits unrestricted use, distribution, reproduction and adaptation in any medium and for any purpose provided that it is properly attributed. For attribution, the original author(s), title, publication source (PeerJ) and either DOI or URL of the article must be cited.
License URL: https://creativecommons.org/licenses/by/4.0/

Keywords: Credible intervals, Highest posterior density intervals, Jeffrey’s rule, Uniform priors, Fiducial quantities, Chiang Mai, Simulation, Rainfall data

Funding: National Science, Research, and Innovation Fund (NSRF) and King Mongkut’s University of Technology North Bangkok KMUTNB-FF-66-03 This research received financial support from the National Science, Research, and Innovation Fund (NSRF) and King Mongkut’s University of Technology North Bangkok (Grant No. KMUTNB-FF-66-03). The funders had no role in study design, data collection and analysis, decision to publish, or preparation of the manuscript.

==============================
Precipitation and flood forecasting are difficult due to rainfall variability. The mean of a delta-gamma distribution can be used to analyze rainfall data for predicting future rainfall, thereby reducing the risks of future disasters due to excessive or too little rainfall. In this study, we construct credible and highest posterior density (HPD) intervals for the mean and the difference between the means of delta-gamma distributions by using Bayesian methods based on Jeffrey’s rule and uniform priors along with a confidence interval based on fiducial quantities. The results of a simulation study indicate that the Bayesian HPD interval based on Jeffrey’s rule prior performed well in terms of coverage probability and provided the shortest expected length. Rainfall data from Chiang Mai province, Thailand, are also used to illustrate the efficacies of the proposed methods.

Introduction

Weather conditions can vary immensely each day and forecasting it accurately for up to 7 days in advance is greatly desired. The climate in a given area provides a broad picture of temperature and rainfall variation over time and is categorized into seasons. For example, Chiang Mai, a province in Northern Thailand, has three seasons: summer (from March to June), the rainy season (from July to October), and winter (from November to February). The major economic output from Chiang Mai is from agriculture, for which rainfall is essential: insufficient or nonexistent rainfall (drought conditions) causes crops to die, whereas excessive rainfall (flooding) destroys crops and can cause disasters such as landslides. Therefore, predicting the amount of rainfall during each period is very important because it would enable farmers to plan the proper use of water resources accordingly. Thus, assessing rainfall dispersion in specific areas by using statistical methods such as the mean is of great importance. Chiang Mai province has an average rainfall of approximately 1,134 mm per year, with the highest rainfall in a day being 166.5 mm (August 14, 1968). The rainiest month is August and the least rainy month is January (Amatayakul & Chomtha, 2013). There can be zero millimeters of rainfall in a month, and so a monthly rainfall series often includes zero values. When a rainwater series only contains positive values, they can be fitted to standard continuous probability distributions such as a gamma distribution. For instance, Sangnawakij & Niwitpong (2017) and Krishnmoorthy, Mathew & Mukherjee (2008) constructed confidence interval for a gamma distribution of monthly rainfall data. However, a delta-gamma distribution (or a zero-inflated gamma distribution) is more suitable for data containing both positive and zero observations. The positive values comprise a gamma distribution with shape and rate parameters while the zero values follow a binomial distribution with proportion of zeros. Inference from a delta-gamma distribution applied to real data has been conducted in many fields. For instance, the testing of body armor for stab resistance in engineering during which a zero value was recorded when the armor was not pierced (Zimmer, Park & Mathew, 2020) and ecological data for biomasses that often contain a high proportion of zeros with skewed positive values (Lecomte et al., 2013).

The confidence interval is a range of observed values within which an unknown population parameter value such as the population mean is known to reside, and a specific confidence level is applied to conclude that the estimated interval contains the true value of the parameter. As an example, Zimmer, Park & Mathew (2020) estimated the coverage probabilities of the 95% upper confidence limits of a zero-inflated gamma distribution for confidence intervals constructed via bias-corrected and accelerated bootstrapping and the bootstrap-calibrated delta method.

The mean of a statistical distribution with a continuous random variable (also known as the expected value or expectation) is the long-running average value of random variables obtained by integrating the product of the variable with its probability defined by the distribution. Since the mean is the most popular measure of central tendency, we are interested in constructing confidence intervals for estimating the population mean as well as expanding the concept to analyze the difference between the means of two populations. Several studies have investigated methods of constructing confidence intervals for functions of the mean. Muralidharan & Kale (2002) used the maximum likelihood concept to estimate the parameters and construct confidence intervals for the mean of a zero-inflated gamma mean population. Ren, Liu & Pu (2021) used fiducial methods to establish simultaneous confidence intervals for the mean of multiple zero-inflated gamma distributions. Thangjai, Niwitpong & Niwitpong (2017) proposed confidence intervals for the mean and the difference between the means of two normal distributions with unknown coefficients of variation. Niwitpong, Koonprasert & Niwitpong (2012) provided confidence intervals for the difference between normal population means with known coefficients of variation. Maneerat, Niwitpong & Niwitpong (2019a) proposed Bayesian methods to construct highest posterior density (HPD) intervals for the mean and the difference between the means of two delta-lognormal distributions. Maneerat, Niwitpong & Niwitpong (2019b) proposed confidence intervals for the mean of a delta-lognormal distribution using the generalized confidence interval and the method of variance estimate recovery based on a weighted beta distribution and variance stabilizing transformation, respectively. Nonetheless, no publications have yet been forthcoming on constructing confidence intervals for the mean and the difference between the means of two delta-gamma distributions.

Herein, we propose confidence intervals for both the mean and the difference between the means of delta-gamma populations. We propose five methods comprising Bayesian credible and HPD intervals based on the Jeffrey’s rule and uniform priors along with a confidence interval based on FQs. The performances of the proposed confidence intervals were evaluated using coverage probabilities and expected lengths via Monte Carlo simulations and were then applied to estimate monthly rainfall data from Chiang Mai province, Thailand, as a demonstration of their efficacy.

In this article, we propose the confidence intervals for the mean of delta-gamma distribution and we expanded to establish confidence intervals for the difference between delta-gamma means are presented in the ‘Methods’ section. The details of the simulation study and the performances of the methods were compared in terms of their coverage probabilities and expected lengths are included in the ‘Results and Discussion’ section. An empirical application of the proposed methods with a monthly rainfall data from Chiang Mai province Thailand are reported in ‘An empirical application’. The last section contains ‘Conclusions’.

Methods

Confidence intervals for the mean of a single delta-gamma distribution

Let X=(X1,X2,...,Xn) be independent and identically distributed random sample from a delta-gamma distribution denoted as X∼DG(α, β, δ). The distribution function of X is given by

(1) F(x;α,β,δ)=(δ;x=0δ+(1−δ)G(x;α,β);x>1,

where G(x;α,β) is a gamma distribution with shape parameter α and rate parameter β and δ=P(x=0) follow binomial distribution n(0)∼bi(n,δ). The maximum likelihood estimator of δ is δ^=n(0)n;n=n(0)+n(1), where n(0) and n(1) are the number of zero and positive observed values, respectively. The population mean of X is τ=(1−δ)αβ, and so the sample mean for τ is τ^=(1−δ^)α^β^.

Krishnmoorthy, Mathew & Mukherjee (2008) showed that data can be transformed using the cube root approximation to develop inferential procedures for a gamma distribution. Suppose that G=(G1,G2,...,Gn) be independent and identically distributed random variables from a gamma distribution, denoted as G(α,β), and that Y = G1/3∼ N(μ, σ2) is approximately normal with μ and σ2 given by Krishnmoorthy & Wang (2016) μ=(αβ)1/3(1−19α) and σ2=19α1/3β2/3, respectively. Since the mean of a gamma distribution M=αβ, then we can rewrite μ and σ2 to obtain

(2) μ=M1/3(1−19βM)andσ2=19βM1/3.

By solving the set of equations in Eq. (2) for M, we obtain

(3) M=(μ2+μ24+σ2)3.

Thus, the mean of a delta-gamma distribution is τ=(1−δ)(μ2+μ24+σ2)3.

Bayesian methods

Bayesian statistical methods use Bayes’ theorem to explain the conditional probability based on the prior distribution of the data. Hence, for the posterior (or conditional) probability of θ given sample x and prior p(θ), the likelihood function p(x|θ) can be defined as

(4) p(θ|x)∝p(θ)p(x|θ).

For non-multimodal densities when p(θ|x) is not symmetric, Box & Tiao (1973) defined the HPD interval as follows.

Suppose p(θ|x) is a posterior distribution, then region R in the parameter space of θ is called the HPD region of content (1 − γ) if the following two conditions are satisfied: Pr(θ ∈ R|x) = 1 − γ.

For θ1 ∈ R and θ2∉R,p(θ1|x)≥p(θ2|x).

As stated earlier, a delta-gamma distribution is a combination of gamma and binomial distributions. Xi⧸=0;i=1,2,...,n(1) following a gamma distribution can be transformed using cubed roots to a normal distribution denoted as Y∼N(μ, σ2). Suppose that Y=(Y1,Y2,...,Yn) be independent and identically distributed random variables with probability density function, the likelihood function is p(y|λ)∝(σ2)−n(1)/2exp(−12σ2∑i=1n(1)(yi−μ)2) and parameter λ = (μ, σ2). Thus, the Fisher information for λ can be obtained from the above equation as

(5) I(λ)=diag[n(1)σ2n(1)2σ4].

The delta-gamma distribution for three unknown parameters is denoted as θ = (μ, σ2, δ) with likelihood function

(6) p(x|θ)∝δn(0)(1−δ)n(1)∏i=1n(1)(σ2)−1/2(−12σ2(xi1/3−μ)2).

Therefore, the Fisher information for θ becomes

(7) I(θ)=diag[nδ(1−δ)n(1)σ2n(1)2σ4].

In the following subsections, we cover the Jeffrey’s rule and uniform priors used to construct Bayesian credible and HPD intervals for the mean of a delta-gamma population.

The Jeffrey’s rule prior is defined as the square root of Fisher information p(θ)=I(θ). Harvey & Van Der Merwe (2012) introduced the Jeffrey’s rule prior for δ in a binomial distribution p(δ) ∝ δ−1/2(1 − δ)1/2 and p(σ2) ∝ 1/σ3. Thus, the joint posterior density of θ is defined as

(8) p(θ|x)=1Beta(n(0)+12,n(1)+32)δn(0)−12(1−δ)n(1)+12×n(1)2πσ2exp(−n(1)2σ2(μ−μ^)2)×(xn(1)2)n(1)2Γ(n(1)2)(σ2)−1−n(1)2exp(−xn(1)2σ2),

where μ^=1n(1)∑i=1n(1)xi1/3 and xn(1)=∑i=1n(1)(xi1/3−μ^)2.

The marginal posteriors of μ and σ2 for the Jeffrey’s rule prior are from normal distribution (μ|σ2,x)J∼N(μ^,σ2/n(1)) and inverse gamma distribution (σ2|x)J∼IG(n(1)/2,xn(1)/2), respectively. In addition, the marginal posterior of δ is beta distribution (δ|x)J∼Beta(n(0)+12,n(1)+32). Thus, the Bayesian credible interval based on the Jeffrey’s rule prior for the mean of a delta-gamma population is obtained as

(9) τJ=(1−δ)M=(1−(δ|x)J)((μ|σ2,x)J2+(μ|σ2,x)J24+(σ2|x)J)3.

Therefore, the 100(1 − γ)% two-sided interval for τ is CIJ=[LJ,UJ]=[τJ(γ/2),τJ(1−γ/2)] where τJ(γ) denotes the (γ)100th percentile of τJ.

The HPD interval has the property that every point within its region has a higher probability than any point outside of it (Noyan & Pham-Gia, 1993; Chen & Shao, 1999). Thus, to find the 100(1 − γ)% HPD interval of τJ, we computed CIHPD.J=[LHPD.J,UHPD.J]=[τHPD.J(γ/2),τHPD.J(1−γ/2)] by using the HPDinterval package in the R software suite, defined by Box & Tiao (1973).

Algorithm 1    • Step 1 generate x from DG(α, β, δ);	
   • Step 2 compute x1/3;	
   • Step 3 generate (μ|σ2,x)J, (σ2|x)J, and (δ|x)J;	
   • Step 4 compute τJ;	
   • Step 5 repeat Step (3) and (4) 5,000 times;	
   • Step 6 compute the 95% Bayesian credible, as given in Eq. (9) from CIJ and HPD interval for τ from CIHPD.J, defined by Box & Tiao (1973);	
   • Step 7 repeat Step (1)–(6) 15,000 times to compute the coverage probabilities and the expected lengths.	

The uniform prior has a distribution that adds no information to the Bayesian inference. According to Bolstad & Curran (2016), the uniform prior for δ of a binomial distribution is p(δ) ∝ 1 while the priors for the mean and variance of a normal distribution are p(μ) ∝ 1 and p(σ2) ∝ 1, respectively. Thus, the joint posterior density function for the uniform prior is defined as

(10) p(θ|x)=1Beta(n(0)+1,n(1)+1)δn(0)(1−δ)n(1)×n(1)2πσ2exp(−n(1)2σ2(μ−μ^)2)×(xn(1)2)n(1)−32Γ(n(1)−32)(σ2)−1−n(1)−32exp(−xn(1)2σ2),

where μ^=1n(1)∑i=1n(1)xi1/3 and xn(1)=∑i=1n(1)(xi1/3−μ^)2.

The marginal posteriors of μ and σ2 for the uniform prior are (μ|σ2,x)U∼N(μ^,σ2/n(1)) for a normal distribution and (σ2|x)U∼IG((n(1)−3)/2,xn(1)/2) for an inverse gamma distribution. In addition, the marginal posterior of δ is beta distribution (δ|x)U∼Beta(n(0)+1,n(1)+1). Thus, the Bayesian credible interval for the mean based on the uniform prior is obtained as

(11) τU=(1−δ)M=(1−(δ|x)U)((μ|σ2,x)U2+(μ|σ2,x)U24+(σ2|x)U)3.

Therefore, the 100(1 − γ)% two-sided credible interval for τ is CIU=[LU,UU]=[τU(γ/2),τU(1−γ/2)] and the 100(1 − γ)% HPD interval for τU is CIHPD.U=[LHPD.U,UHPD.U]=[τHPD.U(γ/2),τHPD.U(1−γ/2)].

Algorithm 2    • Step 1 generate x from DG(α, β, δ);	
   • Step 2 compute x1/3;	
   • Step 3 generate (μ|σ2,x)U, (σ2|x)U, and (δ|x)U;	
   • Step 4 compute τU;	
   • Step 5 repeat Step (3) and (4) 5,000 times;	
   • Step 6 compute the 95% Bayesian credible and HPD interval for τ from CIU and CIHPD.U;	
   • Step 7 repeat Step (1)–(6) 15,000 times to compute the coverage probabilities and the expected lengths.	

Fiducial quantities

After Fisher (1930) introduced fiducial inference, many researchers have used it to establish confidence limits (e.g., Krishnmoorthy & Wang, 2016; Yosboonruang, Niwitpong & Niwitpong, 2019). The fiducial quantity concept depends on the fiducial generalized pivotal quantity defined by Hannig, Iyer & Patterson (2006).

Suppose that X=(X1,X2,...,Xn) be independent and identically distributed random variables with fX(x, ϑ, ζ), where ϑ is the parameter of interest and ζ is a nuisance parameter. Thus, the percentile of generalized pivotal quantity R(X; x, ϑ, ζ) is only a function of ϑ (a fiducial quantity) if it satisfies the following conditions: Given X, the R(X; x, ϑ, ζ) distribution is free of all parameters.

∀x ∈ R+, the observed R(X; x, ϑ, ζ) = ϑ.

From Y∼N(μ, σ2), the sample mean and variance of Y are Y¯≈μ+Zσn(1) and S2≈σ2χn(1)−12, respectively, where Z and χn(1)−1 are standard normal and Chi-squared distributions with n(1)−1 degrees of freedom, respectively.

By estimating μ and σ2 from the sample mean and variance, respectively, and replacing (Y¯,S) with (y¯,s), the respective FQs for μ and σ2 become

(12) Fμ=y¯+Zχn(1)−12(n(1)−1)s2n(1)andFσ2=(n(1)−1)s2χn(1)−12.

Hence, the FQs for the gamma mean are given by

(13) FM=(Fμ2+(Fμ2)2+Fσ2)3.

Yosboonruang, Niwitpong & Niwitpong (2019) used the FQs for 1 − δ to obtain

(14) F1−δ∼12Beta(n(1),n(0)+1)+12Beta(n(1)+1,n(0)).

Thus, the population mean of X is (1 − δ)M and the FQs for a delta-gamma mean are Fτ=(F1−δ)FM. Therefore, the 100(1 − γ)% two-sided interval of the FQs for τ is CIFQ=[LFQ,UFQ]=[Fτ(γ/2),Fτ(1−γ/2)], where Fτ(γ) denotes the (γ)100th percentiles of Fτ.

Algorithm 3    • Step 1 generate x from DG(α, β, δ);	
   • Step 2 compute x1/3;	
   • Step 3 generate Z and χn(1)−12;	
   • Step 4 compute Fμ, Fσ2, F1−δ, FM, and Fτ;	
   • Step 5 repeat Step (4) 5,000 times;	
   • Step 6 compute the 95% confidence interval for τ from CIFQ;	
   • Step 7 repeat Step (1)–(6) 15,000 times to compute the coverage probabilities and the expected lengths.	

Confidence intervals for the difference between delta-gamma means

In this section, we extend the ideas for the single delta-gamma mean confidence intervals to create new ones for the difference between two delta-gamma means. Let X=(X1,X2,...,Xn) and V=(V1,V2,...,Vn) be independent and identically distributed random samples from two delta-gamma distributions denoted as X∼DG(α, β, δ) and V∼DG(α2, β2, δ2), then the difference between their means is simply

(15) ψ=τ−τ2=(1−δ)M−(1−δ2)M2,

where M2=(μ22+μ224+σ22)3. The maximum likelihood estimator of δ2 is δ^2=m(0)m;m=m(0)+m(1), where m(0) and m(1) are the number of zero and positive observed values, respectively. Thus, confidence intervals for parameter ψ can be constructed by using the FQs and Bayesian credible and HPD interval based on the Jeffrey’s rule and uniform priors as follows.

The Bayesian methods

Xi⧸=0;i=1,2,...,n(1) and Vj⧸=0;j=1,2,...,m(1) from gamma distributions can be transformed to normal distributions denoted as Y∼N(μ, σ2) and Y2∼N(μ2,σ22), respectively, by using the cube roots of the data. Thus, the likelihood function is

(16) p(y,y2|φ)∝(σ2)−n(1)/2exp(−12σ2∑i=1n(1)(yi−μ)2)×(σ22)−m(1)/2exp(−12σ22∑j=1m(1)(y2j−μ2)2),

with parameter φ=(μ,σ2,μ2,σ22). Thus, the Fisher information for φ can be derived from the above equation as

(17) I(φ)=diag[n(1)σ2n(1)2σ4m(1)σ22m(1)2σ24].

We can apply the difference between the two independent means as the unknown parameter denoted as ϕ=(μ,σ2,δ,μ2,σ22,δ2) with likelihood function

(18) p(x,v|ϕ)∝δn(0)(1−δ)n(1)∏i=1n(1)(σ2)−1/2(−12σ2(xi1/3−μ)2)×δ2m(0)(1−δ2)m(1)∏j=1m(1)(σ22)−1/2(−12σ22(vj1/3−μ2)2).

Therefore, the Fisher information for ϕ becomes

(19) I(ϕ)=diag[nδ(1−δ)n(1)σ2n(1)2σ4mδ2(1−δ2)m(1)σ22m(1)2σ24].

This can be used to construct confidence intervals for the difference between the means of two delta-gamma populations using Bayesian credible and HPD intervals based on Jeffrey’s rule and uniform priors as follows.

The Jeffrey’s rule prior for δ and δ2 in a binomial distribution comprises p(δ) ∝ δ−1/2 (1 − δ)1/2, p(δ2)∝δ2−1/2(1−δ2)1/2, p(σ2) ∝ 1/σ3 and p(σ22)∝1/σ23. Hence, the joint posterior density of ϕ is defined as

(20) p(ϕ|x,v)=p(θ|x)×1Beta(m(0)+12,m(1)+32)δ2m(0)−12(1−δ2)m(1)+12×m(1)2πσ22exp(−m(1)2σ22(μ2−μ^2)2)×(vm(1)2)m(1)2Γ(m(1)2)(σ22)−1−m(1)2exp(−vm(1)2σ22),

where μ^2=1m(1)∑j=1m(1)vj1/3 and vm(1)=∑j=1m(1)(vj1/3−μ^2)2.

The marginal posteriors of μ2, σ22 and δ2 for the Jeffrey’s rule prior of the difference between two delta-gamma means are (μ2|σ22,v)J∼N(μ^2,σ22/m(1)), (σ22|v)J∼IG(m(1)/2,vm(1)/2), and (δ2|v)J∼Beta(m(0)+12,m(1)+32), respectively. Thus, the Bayesian credible interval based on the Jeffrey’s rule prior is obtained as

(21) ψJ=(1−δ)M−(1−δ2)M2=(1−(δ|x)J)((μ|σ2,x)J2+(μ|σ2,x)J24+(σ2|x)J)3−(1−(δ2|v)J)((μ2|σ22,v)J2+(μ2|σ22,v)J24+(σ22|v)J)3.

Therefore, the 100(1 − γ)% two-sided credible interval for ψ is CId.J=[Ld.J,Ud.J]=[ψJ(γ/2),ψJ(1−γ/2)] and the 100(1 − γ)% HPD interval of ψJ is CId.HPD.J=[Ld.HPD.J,Ud.HPD.J]=[ψHPD.J(γ/2),ψHPD.J(1−γ/2)].

Algorithm 4    • Step 1 generate x and v from DG(α, β, δ) and DG(α2, β2, δ2);	
   • Step 2 compute x1/3 and v1/3;	
   • Step 3 generate (μ|σ2,x)J, (σ2|x)J, (δ|x)J, (μ2|σ22,v)J, (σ22|v)J, and (δ2|v)J;	
   • Step 4 compute ψJ;	
   • Step 5 repeat Step (3) and (4) 5,000 times;	
   • Step 6 compute the 95% Bayesian credible and HPD interval for ψ from CId.J and CId.HPD.J;	
   • Step 7 repeat Step (1)–(6) 15,000 times to compute the coverage probabilities and the expected lengths.	

The uniform prior for δ and δ2 are from binomial distributions p(δ) ∝ 1 and p(δ2) ∝ 1, respectively. The priors for the means are p(μ) ∝ 1 and p(μ2) ∝ 1 and for the variances are p(σ2) ∝ 1 and p(σ22)∝1. Hence, the joint posterior density function for the uniform prior can be defined as

(22) p(ϕ|x,v)=p(θ|x)×1Beta(m(0)+1,m(1)+1)δ2m(0)(1−δ2)m(1)×m(1)2πσ22exp(−m(1)2σ22(μ2−μ^2)2)×(vm(1)2)m(1)−32Γ(m(1)−32)(σ22)−1−m(1)−32exp(−vm(1)2σ22).

The marginal posteriors of μ2, σ22 and δ2 for the uniform prior are (μ2|σ22,v)U∼N(μ^2,σ22/m(1)), (σ22|v)U∼IG((m(1)−3)/2,vm(1)/2), and (δ2|v)U∼Beta(m(0)+1,m(1)+1). Thus, the Bayesian credible interval based on the uniform prior can be obtained as

(23) ψu=(1−δ)M−(1−δ2)M2=(1−(δ|x)U)((μ|σ2,x)U2+(μ|σ2,x)U24+(σ2|x)U)3−(1−(δ2|v)U)((μ2|σ22,v)U2+(μ2|σ22,v)U24+(σ22|v)U)3.

Therefore, the 100(1 − γ)% two-sided credible interval for ψ is CId.U=[Ld.U,Ud.U]=[ψU(γ/2),ψU(1−γ/2)] and the 100(1 − γ)% HPD interval for ψU is CId.HPD.U=[Ld.HPD.U,Ud.HPD.U]=[ψHPD.U(γ/2),ψHPD.U(1−γ/2)].

Algorithm 5    • Step 1 generate x and v from DG(α, β, δ) and DG(α2, β2, δ2);	
   • Step 2 compute x1/3 and v1/3;	
   • Step 3 generate (μ|σ2,x)U, (σ2|x)U, (δ|x)U, (μ2|σ22,v)U, (σ22|v)U, and (δ2|v)U;	
   • Step 4 compute ψU;	
   • Step 5 repeat Step (3) and (4) 5,000 times;	
   • Step 6 compute the 95% Bayesian credible and HPD interval for ψ from CId.U and CId.HPD.U;	
   • Step 7 repeat Step (1)–(6) 15,000 times to compute the coverage probabilities and the expected lengths.	

FQs

The FQs for μ2 and σ22 are given by Fμ2=y¯2+Z2χm(1)−12(m(1)−1)s2m(1) and Fσ22=(m(1)−1)s2χm(1)−12, respectively where Z2 and χm(1)−12 are standard normal and Chi-squared distributions with m(1) − 1 degrees of freedom, respectively. Hence, the FQs for 1 − δ2 and M2 can respectively be obtained as

(24) F1−δ2∼12Beta(m(1),m(0)+1)+12Beta(m(1)+1,m(0)),

and

(25) FM2=(Fμ22+(Fμ22)2+Fσ22)3.

Thus, the FQs for the difference between two delta-gamma means can be derived as

(26) Fψ=Fτ−Fτ2=(F1−δ)FM−(F1−δ2)FM2.

Therefore, the 100(1 − γ)% two-sided confidence interval using the FQs for ψ is CId.FQ=[Ld.FQ,Ud.FQ]=[Fψ(γ/2),Fψ(1−γ/2)], where Fψ(γ) denotes the (γ)100th percentiles of Fψ.

Algorithm 6    • Step 1 generate x and v from DG(α, β, δ) and DG(α2, β2, δ2);	
   • Step 2 compute x1/3 and v1/3;	
   • Step 3 generate Z, Z2, χn(1)−12 and χm(1)−12;	
   • Step 4 compute Fμ, Fμ2, Fσ2, Fσ22, F1−δ, F1−δ2, FM, FM2, and Fψ;	
   • Step 5 repeat Step (4) 5,000 times;	
   • Step 6 compute the 95% confidence interval for ψ from CId.FQ;	
   • Step 7 repeat Step (1)–(6) 15,000 times to compute the coverage probabilities and the expected lengths.	

Simulation studies and results

The five methods for establishing new confidence intervals for the mean and the difference between the means of two delta-gamma distributions were tested via a Monte Carlo simulation study conducted using the R statistical program (R Core Team, 2021). The performances of the five proposed methods were compared in terms of their coverage probabilities and expected lengths respectively derived as

(27) CP=c(L≤τorψ≤U)15,000andEL=∑k=115,000(Uk−Lk)15,000,

where c(L≤τorψ≤U) is the number of simulation runs for τ or ψ. The simulation results are presented for significance level γ = 0.05. The best-performing confidence interval was chosen with a coverage probability greater than or close to the nominal confidence level of 0.95 and the shortest expected length.

For the single delta-gamma mean, the data were generated for X∼DG(α, β, δ) with sample sizes n = 30, 50, 100, or 200 and the probability of zeros δ = 0.2, 0.5, or 0.7, for which we set α = 5.5 or 6.0, 2.0 or 2.5, and 1.25 or 1.5, respectively. Last, we set β = 1.0 or 2.0 for all cases. Subsequently, the performances of the confidence intervals at the nominal confidence level of 95% for τ were computed.

For the difference between two delta-gamma means, the data were generated for two independent delta-gamma distributions, X∼DG(α, β, δ) and V∼DG(α2, β2, δ2). For equal sample sizes (n = m), we used (30, 30), (50, 50), (100, 100), or (200, 200), and for unequal sample sizes (n ≠ m), we used (30, 50), (50, 100), or (100, 200). For probabilities of zeros (δ, δ2) = (0.2, 0.2), we set (α, β) and (α2, β2) as (5.5, 1.0), (5.5, 2.0), (6.0, 1.0), or (6.0, 2.0); for (δ, δ2) = (0.5, 0.5), we set (α, β) and (α2, β2) as (2.0, 1.0), (2.0, 2.0), (2.5, 1.0), or (2.5, 2.0); and for (δ, δ2) = (0.7, 0.7), we set (α, β) and (α2, β2) as (1.25, 1.0), (1.25, 2.0), (1.5, 1.0), or (1.5, 2.0). For all of the simulations, the number of replications was set as 15,000 with 5,000 repetitions.

The coverage probabilities and expected lengths of the 95% confidence intervals for τ are reported in Table 1. It can be seen that the coverage probabilities of the Bayesian HPD interval based on the uniform prior and the FQ confidence interval were greater than or close to the nominal confidence level of 0.95 in all cases where as those of the Bayesian credible and HPD intervals based on the Jeffrey’s rule prior and the Bayesian credible interval based on the uniform prior were less than the nominal confidence level for some cases. However, the expected lengths of the Bayesian HPD interval based on the Jeffrey’s rule prior were shorter than the other methods as shown in Fig. 1. Therefore, the Bayesian HPD interval based on the Jeffrey’s rule prior is recommended for constructing the confidence interval for the mean of a single delta-gamma distribution. The coverage probabilities and expected lengths of the 95% two-sided confidence interval for ψ with equal and unequal sample sizes are listed in Tables 2 and 3, respectively. The results show that the Bayesian HPD interval based on the Jeffrey’s rule prior, the Bayesian credible and HPD intervals based on the uniform prior, and the FQ confidence interval provided coverage probabilities that were greater than or close to the nominal confidence level of 0.95 in all cases whereas the Bayesian credible interval on the Jeffrey’s rule prior with (δ, δ2) = (0.7, 0.7) provided ones that were less than the nominal confidence level for some cases. Since the expected lengths of the Bayesian HPD interval based on the Jeffrey’s rule prior were the shortest as shown in Figs. 2 and 3. Thus, we can recommend it for constructing the confidence interval for the difference between the means of two delta-gamma distributions with equal and unequal sample sizes. Furthermore, the results for the difference between the means of two delta-gamma distributions for sample size n > m yielded similar results to those for n < m.

Figure 1 Graphs of (A) coverage probability and (B) expected length of the proposed methods for the mean of a delta-gamma distribution.

Figure 2 Graphs of (A) coverage probability and (B) expected length of the proposed methods for the difference between the means of delta-gamma distributions with equal sample sizes.

Figure 3 Graphs of (A) coverage probability and (B) expected length of the proposed methods for the difference between the means of delta-gamma distributions with unequal sample sizes.

Table 1 Coverage probabilities and expected lengths of the proposed methods for the 95% confidence intervals for the mean of a delta-gamma distribution.

n	δ	(α, β)	Coverage probability (Expected length)	
			CI J	CI HPD.J	CI U	CI HPD.U	CI FQ	
30	0.2	(5.5, 1.0)	0.9441	0.9480	0.9454	0.9524	0.9842	
			(1.5459)	(1.5247)	(1.6236)	(1.6064)	(1.9300)	
		(5.5, 2.0)	0.9451	0.9488	0.9468	0.9532	0.9842	
			(0.7760)	(0.7657)	(0.8146)	(0.8062)	(0.9650)	
		(6.0, 1.0)	0.9494	0.9562	0.9508	0.9583	0.9857	
			(1.6776)	(1.6530)	(1.7575)	(1.7377)	(2.0448)	
		(6.0, 2.0)	0.9508	0.9569	0.9522	0.9588	0.9857	
			(0.8418)	(0.8299)	(0.8815)	(0.8718)	(1.0224)	
	0.5	(2.0, 1.0)	0.9552	0.9528	0.9700	0.9695	0.9830	
			(0.7596)	(0.7531)	(0.8553)	(0.8392)	(0.9711)	
		(2.0, 2.0)	0.9570	0.9550	0.9714	0.9705	0.9830	
			(0.3829)	(0.3795)	(0.4303)	(0.4221)	(0.4855)	
		(2.5, 1.0)	0.9690	0.9682	0.9796	0.9790	0.9873	
			(0.9209)	(0.9150)	(1.0102)	(0.9972)	(1.1159)	
		(2.5, 2.0)	0.9703	0.9696	0.9805	0.9797	0.9873	
			(0.4634)	(0.4604)	(0.5077)	(0.5012)	(0.5579)	
	0.7	(1.25, 1.0)	0.9455	0.9496	0.9643	0.9832	0.9784	
			(0.5091)	(0.4872)	(0.8029)	(0.7002)	(0.6961)	
		(1.25, 2.0)	0.9497	0.9531	0.9656	0.9841	0.9784	
			(0.2583)	(0.2471)	(0.4039)	(0.3526)	(0.3480)	
		(1.5, 1.0)	0.9585	0.9599	0.9758	0.9870	0.9829	
			(0.5814)	(0.5615)	(0.8665)	(0.7708)	(0.7654)	
		(1.5, 2.0)	0.9609	0.9623	0.9770	0.9877	0.9829	
			(0.2945)	(0.2843)	(0.4359)	(0.3879)	(0.3827)	
50	0.2	(5.5, 1.0)	0.9485	0.9512	0.9484	0.9524	0.9838	
			(1.2119)	(1.1988)	(1.2452)	(1.2333)	(1.4736)	
		(5.5, 2.0)	0.9494	0.9520	0.9492	0.9533	0.9838	
			(0.6072)	(0.6008)	(0.6239)	(0.6179)	(0.7368)	
		(6.0, 1.0)	0.9582	0.9607	0.9561	0.9604	0.9856	
			(1.3166)	(1.3017)	(1.3512)	(1.3376)	(1.5606)	
		(6.0, 2.0)	0.9585	0.9620	0.9564	0.9606	0.9856	
			(0.6596)	(0.6523)	(0.6768)	(0.6701)	(0.7803)	
	0.5	(2.0, 1.0)	0.9580	0.9568	0.9649	0.9653	0.9840	
			(0.5840)	(0.5807)	(0.6138)	(0.6093)	(0.7172)	
		(2.0, 2.0)	0.9583	0.9571	0.9656	0.9650	0.9840	
			(0.2933)	(0.2917)	(0.3082)	(0.3059)	(0.3586)	
		(2.5, 1.0)	0.9694	0.9683	0.9754	0.9748	0.9865	
			(0.7109)	(0.7076)	(0.7390)	(0.7347)	(0.8283)	
		(2.5, 2.0)	0.9706	0.9691	0.9760	0.9746	0.9865	
			(0.3568)	(0.3551)	(0.3707)	(0.3685)	(0.4141)	
	0.7	(1.25, 1.0)	0.9467	0.9480	0.9597	0.9676	0.9780	
			(0.3689)	(0.3609)	(0.4307)	(0.4137)	(0.4685)	
		(1.25, 2.0)	0.9490	0.9503	0.9612	0.9692	0.9780	
			(0.1861)	(0.1820)	(0.2167)	(0.2082)	(0.2342)	
		(1.5, 1.0)	0.9610	0.9634	0.9718	0.9762	0.9833	
			(0.4270)	(0.4195)	(0.4863)	(0.4709)	(0.5226)	
		(1.5, 2.0)	0.9630	0.9634	0.9725	0.9769	0.9833	
			(0.2151)	(0.2113)	(0.2445)	(0.2368)	(0.2613)	
100	0.2	(5.5, 1.0)	0.9560	0.9569	0.9535	0.9567	0.9853	
			(0.8663)	(0.8595)	(0.8779)	(0.8714)	(1.0319)	
		(5.5, 2.0)	0.9562	0.9572	0.9536	0.9572	0.9853	
			(0.4336)	(0.4302)	(0.4394)	(0.4361)	(0.5159)	
		(6.0, 1.0)	0.9638	0.9643	0.9618	0.9636	0.9850	
			(0.9421)	(0.9346)	(0.9539)	(0.9467)	(1.0945)	
		(6.0, 2.0)	0.9640	0.9650	0.9618	0.9637	0.9850	
			(0.4715)	(0.4677)	(0.4774)	(0.4738)	(0.5472)	
	0.5	(2.0, 1.0)	0.9586	0.9576	0.9614	0.9606	0.9830	
			(0.4106)	(0.4088)	(0.4189)	(0.4169)	(0.4911)	
		(2.0, 2.0)	0.9588	0.9569	0.9618	0.9611	0.9830	
			(0.2057)	(0.2048)	(0.2099)	(0.2088)	(0.2455)	
		(2.5, 1.0)	0.9724	0.9724	0.9742	0.9732	0.9855	
			(0.5027)	(0.5006)	(0.5104)	(0.5081)	(0.5695)	
		(2.5, 2.0)	0.9726	0.9726	0.9744	0.9735	0.9855	
			(0.2518)	(0.2507)	(0.2556)	(0.2545)	(0.2847)	
	0.7	(1.25, 1.0)	0.9482	0.9476	0.9553	0.9576	0.9790	
			(0.2499)	(0.2472)	(0.2632)	(0.2597)	(0.3045)	
		(1.25, 2.0)	0.9493	0.9476	0.9560	0.9586	0.9790	
			(0.1255)	(0.1241)	(0.1321)	(0.1303)	(0.1522)	
		(1.5, 1.0)	0.9632	0.9635	0.9686	0.9684	0.9836	
			(0.2919)	(0.2892)	(0.3044)	(0.3010)	(0.3423)	
		(1.5, 2.0)	0.9642	0.9642	0.9694	0.9692	0.9836	
			(0.1465)	(0.1451)	(0.1527)	(0.1510)	(0.1711)	
200	0.2	(5.5, 1.0)	0.9603	0.9612	0.9600	0.9620	0.9855	
			(0.6162)	(0.6125)	(0.6201)	(0.6165)	(0.7264)	
		(5.5, 2.0)	0.9602	0.9616	0.9598	0.9617	0.9855	
			(0.3082)	(0.3064)	(0.3102)	(0.3084)	(0.3632)	
		(6.0, 1.0)	0.9676	0.9676	0.9668	0.9678	0.9862	
			(0.6702)	(0.6661)	(0.6745)	(0.6705)	(0.7711)	
		(6.0, 2.0)	0.9678	0.9680	0.9668	0.9676	0.9862	
			(0.3352)	(0.3332)	(0.3374)	(0.3354)	(0.3855)	
	0.5	(2.0, 1.0)	0.9601	0.9591	0.9616	0.9607	0.9836	
			(0.2898)	(0.2886)	(0.2924)	(0.2911)	(0.3419)	
		(2.0, 2.0)	0.9604	0.9594	0.9616	0.9607	0.9836	
			(0.1450)	(0.1444)	(0.1463)	(0.1457)	(0.1709)	
		(2.5, 1.0)	0.9747	0.9736	0.9751	0.9746	0.9865	
			(0.3561)	(0.3546)	(0.3585)	(0.3570)	(0.3980)	
		(2.5, 2.0)	0.9748	0.9739	0.9752	0.9751	0.9865	
			(0.1782)	(0.1774)	(0.1794)	(0.1787)	(0.1990)	
	0.7	(1.25, 1.0)	0.9526	0.9507	0.9551	0.9551	0.9792	
			(0.1731)	(0.1719)	(0.1769)	(0.1755)	(0.2069)	
		(1.25, 2.0)	0.9531	0.9514	0.9556	0.9568	0.9792	
			(0.0867)	(0.0861)	(0.0886)	(0.0879)	(0.1034)	
		(1.5, 1.0)	0.9658	0.9650	0.9678	0.9672	0.9840	
			(0.2036)	(0.2023)	(0.2071)	(0.2057)	(0.2343)	
		(1.5, 2.0)	0.9662	0.9651	0.9682	0.9678	0.9840	
			(0.1020)	(0.1013)	(0.1037)	(0.1030)	(0.1171)	

Table 2 Coverage probabilities and expected lengths of the 95% confidence intervals for the difference between the means of two delta-gamma distributions (n = m).

n, m	(δ, δ2)	(α, β)	(α2, β2)	Coverage probability (Expected length)	
				CI d.J	CI d.HPD.J	CI d.U	CI d.HPD.U	CI d.FQ	
(30, 30)	(0.2, 0.2)	(5.5, 2.0)	(5.5, 2.0)	0.9670	0.9629	0.9756	0.9725	0.9878	
				(1.1135)	(1.1087)	(1.1652)	(1.1603)	(1.3725)	
		(6.0, 2.0)	(5.5, 1.0)	0.9586	0.9572	0.9674	0.9667	0.9860	
				(1.7761)	(1.7643)	(1.8603)	(1.8492)	(2.1877)	
		(5.5, 2.0)	(6.0, 1.0)	0.9613	0.9613	0.9662	0.9676	0.9862	
				(1.8634)	(1.8483)	(1.9501)	(1.9364)	(2.2694)	
		(6.0, 1.0)	(6.0, 1.0)	0.9735	0.9692	0.9802	0.9777	0.9899	
				(2.4093)	(2.3990)	(2.5172)	(2.5067)	(2.9070)	
	(0.5, 0.5)	(2.0, 2.0)	(2.0, 2.0)	0.9549	0.9580	0.9691	0.9762	0.9842	
				(0.5474)	(0.5447)	(0.6175)	(0.6132)	(0.7070)	
		(2.5, 2.0)	(2.0, 1.0)	0.9589	0.9610	0.9733	0.9756	0.9843	
				(0.8972)	(0.8919)	(1.0057)	(0.9957)	(1.1439)	
		(2.0, 2.0)	(2.5, 1.0)	0.9660	0.9663	0.9751	0.9764	0.9853	
				(1.0026)	(0.9972)	(1.1055)	(1.0958)	(1.2338)	
		(2.5, 1.0)	(2.5, 1.0)	0.9700	0.9716	0.9784	0.9826	0.9871	
				(1.3149)	(1.3091)	(1.4456)	(1.4374)	(1.6174)	
	(0.7, 0.7)	(1.25, 2.0)	(1.25, 2.0)	0.9486	0.9666	0.9787	0.9928	0.9824	
				(0.3820)	(0.3770)	(0.6247)	(0.6049)	(0.5358)	
		(1.5, 2.0)	(1.25, 1.0)	0.9479	0.9594	0.9763	0.9910	0.9800	
				(0.6061)	(0.5945)	(0.9721)	(0.9253)	(0.8432)	
		(1.25, 2.0)	(1.5, 1.0)	0.9579	0.9649	0.9824	0.9917	0.9834	
				(0.6527)	(0.6404)	(1.0082)	(0.9609)	(0.8841)	
		(1.5, 1.0)	(1.5, 1.0)	0.9590	0.9721	0.9839	0.9939	0.9853	
				(0.8521)	(0.8430)	(1.3168)	(1.2791)	(1.1625)	
(50, 50)	(0.2, 0.2)	(5.5, 2.0)	(5.5, 2.0)	0.9669	0.9633	0.9724	0.9706	0.9871	
				(0.8672)	(0.8636)	(0.8901)	(0.8864)	(1.0457)	
		(6.0, 2.0)	(5.5, 1.0)	0.9618	0.9620	0.9650	0.9652	0.9846	
				(1.3878)	(1.3795)	(1.4251)	(1.4169)	(1.6700)	
		(5.5, 2.0)	(6.0, 1.0)	0.9636	0.9642	0.9662	0.9664	0.9854	
				(1.4569)	(1.4466)	(1.4947)	(1.4850)	(1.7289)	
		(6.0, 1.0)	(6.0, 1.0)	0.9726	0.9704	0.9770	0.9747	0.9870	
				(1.8798)	(1.8720)	(1.9269)	(1.9189)	(2.2142)	
	(0.5, 0.5)	(2.0, 2.0)	(2.0, 2.0)	0.9574	0.9589	0.9651	0.9666	0.9820	
				(0.4173)	(0.4155)	(0.4388)	(0.4369)	(0.5158)	
		(2.5, 2.0)	(2.0, 1.0)	0.9620	0.9618	0.9699	0.9700	0.9854	
				(0.6871)	(0.6840)	(0.7205)	(0.7167)	(0.8374)	
		(2.0, 2.0)	(2.5, 1.0)	0.9680	0.9682	0.9718	0.9722	0.9863	
				(0.7706)	(0.7672)	(0.8024)	(0.7984)	(0.9079)	
		(2.5, 1.0)	(2.5, 1.0)	0.9754	0.9753	0.9804	0.9800	0.9884	
				(1.0093)	(1.0051)	(1.0496)	(1.0452)	(1.1857)	
	(0.7, 0.7)	(1.25, 2.0)	(1.25, 2.0)	0.9454	0.9564	0.9620	0.9752	0.9779	
				(0.2691)	(0.2672)	(0.3163)	(0.3131)	(0.3468)	
		(1.5, 2.0)	(1.25, 1.0)	0.9482	0.9549	0.9653	0.9728	0.9794	
				(0.4325)	(0.4282)	(0.5045)	(0.4961)	(0.5522)	
		(1.25, 2.0)	(1.5, 1.0)	0.9588	0.9638	0.9718	0.9773	0.9833	
				(0.4711)	(0.4660)	(0.5412)	(0.5319)	(0.5873)	
		(1.5, 1.0)	(1.5, 1.0)	0.9630	0.9693	0.9746	0.9828	0.9848	
				(0.6156)	(0.6120)	(0.7057)	(0.6996)	(0.7696)	
(100, 100)	(0.2, 0.2)	(5.5, 2.0)	(5.5, 2.0)	0.9614	0.9590	0.9645	0.9632	0.9838	
				(0.6159)	(0.6134)	(0.6238)	(0.6213)	(0.7299)	
		(6.0, 2.0)	(5.5, 1.0)	0.9604	0.9583	0.9638	0.9622	0.9849	
				(0.9894)	(0.9842)	(1.0019)	(0.9969)	(1.1692)	
		(5.5, 2.0)	(6.0, 1.0)	0.9668	0.9668	0.9684	0.9680	0.9863	
				(1.0394)	(1.0335)	(1.0526)	(1.0468)	(1.2109)	
		(6.0, 1.0)	(6.0, 1.0)	0.9702	0.9688	0.9718	0.9704	0.9863	
				(1.3384)	(1.3330)	(1.3553)	(1.3498)	(1.5500)	
	(0.5, 0.5)	(2.0, 2.0)	(2.0, 2.0)	0.9598	0.9587	0.9634	0.9628	0.9842	
				(0.2921)	(0.2910)	(0.2979)	(0.2967)	(0.3505)	
		(2.5, 2.0)	(2.0, 1.0)	0.9628	0.9622	0.9671	0.9660	0.9842	
				(0.4825)	(0.4805)	(0.4914)	(0.4893)	(0.5703)	
		(2.0, 2.0)	(2.5, 1.0)	0.9720	0.9711	0.9730	0.9730	0.9861	
				(0.5441)	(0.5418)	(0.5527)	(0.5504)	(0.6223)	
		(2.5, 1.0)	(2.5, 1.0)	0.9744	0.9738	0.9766	0.9764	0.9870	
				(0.7129)	(0.7101)	(0.7238)	(0.7208)	(0.8105)	
	(0.7, 0.7)	(1.25, 2.0)	(1.25, 2.0)	0.9508	0.9548	0.9561	0.9608	0.9778	
				(0.1797)	(0.1788)	(0.1894)	(0.1884)	(0.2204)	
		(1.5, 2.0)	(1.25, 1.0)	0.9513	0.9531	0.9580	0.9616	0.9804	
				(0.2918)	(0.2900)	(0.3068)	(0.3046)	(0.3545)	
		(1.25, 2.0)	(1.5, 1.0)	0.9618	0.9626	0.9672	0.9694	0.9834	
				(0.3193)	(0.3172)	(0.3338)	(0.3313)	(0.3788)	
		(1.5, 1.0)	(1.5, 1.0)	0.9600	0.9644	0.9659	0.9692	0.9826	
				(0.4165)	(0.4147)	(0.4350)	(0.4330)	(0.4939)	
(200, 200)	(0.2, 0.2)	(5.5, 2.0)	(5.5, 2.0)	0.9638	0.9622	0.9652	0.9635	0.9858	
				(0.4370)	(0.4352)	(0.4398)	(0.4380)	(0.5142)	
		(6.0, 2.0)	(5.5, 1.0)	0.9649	0.9630	0.9661	0.9651	0.9855	
				(0.7024)	(0.6992)	(0.7071)	(0.7038)	(0.8223)	
		(5.5, 2.0)	(6.0, 1.0)	0.9675	0.9654	0.9682	0.9676	0.9863	
				(0.7387)	(0.7350)	(0.7433)	(0.7396)	(0.8524)	
		(6.0, 1.0)	(6.0, 1.0)	0.9672	0.9656	0.9691	0.9676	0.9840	
				(0.9502)	(0.9463)	(0.9560)	(0.9521)	(1.0910)	
	(0.5, 0.5)	(2.0, 2.0)	(2.0, 2.0)	0.9581	0.9570	0.9603	0.9589	0.9829	
				(0.2054)	(0.2046)	(0.2073)	(0.2064)	(0.2428)	
		(2.5, 2.0)	(2.0, 1.0)	0.9654	0.9645	0.9668	0.9655	0.9854	
				(0.3408)	(0.3395)	(0.3437)	(0.3422)	(0.3969)	
		(2.0, 2.0)	(2.5, 1.0)	0.9721	0.9713	0.9731	0.9723	0.9851	
				(0.3845)	(0.3830)	(0.3872)	(0.3857)	(0.4336)	
		(2.5, 1.0)	(2.5, 1.0)	0.9764	0.9748	0.9768	0.9760	0.9880	
				(0.5048)	(0.5028)	(0.5081)	(0.5061)	(0.5654)	
	(0.7, 0.7)	(1.25, 2.0)	(1.25, 2.0)	0.9526	0.9541	0.9553	0.9563	0.9797	
				(0.1235)	(0.1229)	(0.1261)	(0.1256)	(0.1482)	
		(1.5, 2.0)	(1.25, 1.0)	0.9550	0.9550	0.9572	0.9586	0.9804	
				(0.2021)	(0.2011)	(0.2062)	(0.2051)	(0.2400)	
		(1.25, 2.0)	(1.5, 1.0)	0.9622	0.9619	0.9654	0.9640	0.9836	
				(0.2219)	(0.2208)	(0.2259)	(0.2247)	(0.2576)	
		(1.5, 1.0)	(1.5, 1.0)	0.9676	0.9681	0.9698	0.9698	0.9834	
				(0.2892)	(0.2880)	(0.2942)	(0.2930)	(0.3346)	

Table 3 Coverage probabilities and expected lengths of the 95% confidence intervals for the difference between the means of two delta-gamma distributions (n ≠ m).

n, m	(δ, δ2)	(α, β)	(α2, β2)	Coverage probability (Expected length)	
				CI d.J	CI d.HPD.J	CI d.U	CI d.HPD.U	CI d.FQ	
(30, 50)	(0.2, 0.2)	(5.5, 2.0)	(5.5, 2.0)	0.9638	0.9605	0.9709	0.9682	0.9862	
				(0.9963)	(0.9915)	(1.0353)	(1.0305)	(1.2181)	
		(6.0, 2.0)	(5.5, 1.0)	0.9646	0.9613	0.9708	0.9690	0.9852	
				(1.4907)	(1.4837)	(1.5392)	(1.5320)	(1.8000)	
		(5.5, 2.0)	(6.0, 1.0)	0.9676	0.9658	0.9719	0.9715	0.9866	
				(1.5404)	(1.5318)	(1.5886)	(1.5801)	(1.8400)	
		(6.0, 1.0)	(6.0, 1.0)	0.9704	0.9680	0.9761	0.9748	0.9874	
				(2.1567)	(2.1462)	(2.2383)	(2.2277)	(2.5790)	
	(0.5, 0.5)	(2.0, 2.0)	(2.0, 2.0)	0.9616	0.9633	0.9720	0.9758	0.9855	
				(0.4872)	(0.4847)	(0.5353)	(0.5307)	(0.6183)	
		(2.5, 2.0)	(2.0, 1.0)	0.9616	0.9628	0.9692	0.9722	0.9848	
				(0.7497)	(0.7465)	(0.8026)	(0.7987)	(0.9247)	
		(2.0, 2.0)	(2.5, 1.0)	0.9719	0.9726	0.9775	0.9786	0.9878	
				(0.8133)	(0.8098)	(0.8629)	(0.8589)	(0.9762)	
		(2.5, 1.0)	(2.5, 1.0)	0.9728	0.9733	0.9792	0.9812	0.9876	
				(1.1706)	(1.1653)	(1.2603)	(1.2520)	(1.4129)	
	(0.7, 0.7)	(1.25, 2.0)	(1.25, 2.0)	0.9488	0.9614	0.9719	0.9878	0.9796	
				(0.3276)	(0.3226)	(0.4785)	(0.4504)	(0.4437)	
		(1.5, 2.0)	(1.25, 1.0)	0.9510	0.9615	0.9696	0.9847	0.9811	
				(0.4858)	(0.4817)	(0.6453)	(0.6298)	(0.6407)	
		(1.25, 2.0)	(1.5, 1.0)	0.9572	0.9653	0.9730	0.9852	0.9836	
				(0.5123)	(0.5080)	(0.6670)	(0.6523)	(0.6628)	
		(1.5, 1.0)	(1.5, 1.0)	0.9618	0.9706	0.9789	0.9900	0.9840	
				(0.7373)	(0.7283)	(1.0280)	(0.9754)	(0.9708)	
(50, 100)	(0.2, 0.2)	(5.5, 2.0)	(5.5, 2.0)	0.9626	0.9604	0.9658	0.9642	0.9836	
				(0.7508)	(0.7470)	(0.7671)	(0.7633)	(0.9011)	
		(6.0, 2.0)	(5.5, 1.0)	0.9672	0.9643	0.9715	0.9685	0.9870	
				(1.0951)	(1.0905)	(1.1144)	(1.1097)	(1.2958)	
		(5.5, 2.0)	(6.0, 1.0)	0.9698	0.9694	0.9738	0.9718	0.9890	
				(1.1266)	(1.1215)	(1.1456)	(1.1403)	(1.3217)	
		(6.0, 1.0)	(6.0, 1.0)	0.9670	0.9654	0.9698	0.9689	0.9862	
				(1.6306)	(1.6224)	(1.6654)	(1.6572)	(1.9119)	
	(0.5, 0.5)	(2.0, 2.0)	(2.0, 2.0)	0.9565	0.9570	0.9642	0.9644	0.9822	
				(0.3595)	(0.3579)	(0.3742)	(0.3723)	(0.4391)	
		(2.5, 2.0)	(2.0, 1.0)	0.9656	0.9648	0.9696	0.9703	0.9862	
				(0.5468)	(0.5445)	(0.5623)	(0.5600)	(0.6498)	
		(2.0, 2.0)	(2.5, 1.0)	0.9716	0.9720	0.9740	0.9743	0.9862	
				(0.5845)	(0.5822)	(0.5989)	(0.5964)	(0.6796)	
		(2.5, 1.0)	(2.5, 1.0)	0.9732	0.9732	0.9764	0.9762	0.9858	
				(0.8749)	(0.8712)	(0.9022)	(0.8981)	(1.0145)	
	(0.7, 0.7)	(1.25, 2.0)	(1.25, 2.0)	0.9494	0.9551	0.9611	0.9710	0.9794	
				(0.2278)	(0.2257)	(0.2584)	(0.2539)	(0.2877)	
		(1.5, 2.0)	(1.25, 1.0)	0.9527	0.9592	0.9629	0.9706	0.9810	
				(0.3343)	(0.3326)	(0.3657)	(0.3632)	(0.4128)	
		(1.25, 2.0)	(1.5, 1.0)	0.9579	0.9622	0.9660	0.9723	0.9830	
				(0.3510)	(0.3491)	(0.3807)	(0.3784)	(0.4267)	
		(1.5, 1.0)	(1.5, 1.0)	0.9621	0.9676	0.9706	0.9766	0.9842	
				(0.5244)	(0.5205)	(0.5828)	(0.5748)	(0.6417)	
(100, 200)	(0.2, 0.2)	(5.5, 2.0)	(5.5, 2.0)	0.9633	0.9614	0.9651	0.9632	0.9858	
				(0.5338)	(0.5313)	(0.5396)	(0.5371)	(0.6317)	
		(6.0, 2.0)	(5.5, 1.0)	0.9678	0.9660	0.9694	0.9678	0.9868	
				(0.7782)	(0.7750)	(0.7851)	(0.7819)	(0.9105)	
		(5.5, 2.0)	(6.0, 1.0)	0.9703	0.9688	0.9712	0.9700	0.9874	
				(0.8003)	(0.7969)	(0.8068)	(0.8033)	(0.9285)	
		(6.0, 1.0)	(6.0, 1.0)	0.9697	0.9686	0.9710	0.9686	0.9866	
				(1.1591)	(1.1538)	(1.1711)	(1.1658)	(1.3397)	
	(0.5, 0.5)	(2.0, 2.0)	(2.0, 2.0)	0.9590	0.9582	0.9623	0.9623	0.9838	
				(0.2522)	(0.2512)	(0.2563)	(0.2552)	(0.3007)	
		(2.5, 2.0)	(2.0, 1.0)	0.9658	0.9666	0.9686	0.9678	0.9874	
				(0.3851)	(0.3836)	(0.3895)	(0.3880)	(0.4475)	
		(2.0, 2.0)	(2.5, 1.0)	0.9728	0.9721	0.9751	0.9734	0.9876	
				(0.4117)	(0.4100)	(0.4159)	(0.4142)	(0.4693)	
		(2.5, 1.0)	(2.5, 1.0)	0.9734	0.9726	0.9750	0.9744	0.9865	
				(0.6177)	(0.6151)	(0.6255)	(0.6229)	(0.6984)	
	(0.7, 0.7)	(1.25, 2.0)	(1.25, 2.0)	0.9520	0.9546	0.9578	0.9616	0.9799	
				(0.1540)	(0.1532)	(0.1607)	(0.1596)	(0.1872)	
		(1.5, 2.0)	(1.25, 1.0)	0.9544	0.9552	0.9588	0.9600	0.9805	
				(0.2286)	(0.2276)	(0.2356)	(0.2346)	(0.2726)	
		(1.25, 2.0)	(1.5, 1.0)	0.9582	0.9597	0.9626	0.9637	0.9815	
				(0.2403)	(0.2393)	(0.2469)	(0.2458)	(0.2825)	
		(1.5, 1.0)	(1.5, 1.0)	0.9640	0.9665	0.9678	0.9704	0.9825	
				(0.3577)	(0.3559)	(0.3703)	(0.3681)	(0.4198)	

An empirical application

In this part of the study, we approximate the mean of monthly rainfall data from Chiang Mai province, Thailand (https://www.hydro-1.net/), using the five confidence intervals proposed in this paper to illustrate their efficacies. There are three cases as follows. Case 1 was used to test the mean of a delta-gamma distribution for which we used rainfall data from only one rain station in Chiang Mai to provide a sample size consistent with that used in the simulation study. The difference between the means of two delta-gamma distributions with equal sample sizes was investigated in Case 2 by using rainfall data over a period of time at the same station in Chiang Mai for various months within the same season. For Case 3, we compared the means of two delta-gamma distributions with uneven sample sizes by combining data from several stations in Chiang Mai for the same month.

Example 1: testing the mean of a single delta-gamma distribution

Rainfall data were obtained from the Upper Northern Region Irrigation Hydrology Center (2021). We used monthly rainfall data (mm) from Irrigation Office Station I, Chiang Mai city, comprising 50 observations in January from 1972 to 2021. The densities for the rainfall data are shown in Fig. 4.

Figure 4 The densities of the rainfall data from Irrigation Office Station I, Chiang Mai city.

Next, we tested the distributions of positive rainfall datasets using the minimum Akaike information criterion (AIC) defined as

(28) AIC=−2ln⁡L+2k,

where L is the likelihood function and k is the number of parameters. The results in Table 4 show that the positive rainfall dataset for Irrigation Office Station I fit a gamma distribution, as confirmed by the AIC because the AIC value for this distribution was the smallest. The Q–Q plots of positive rainfall data showing that they follow gamma distributions are exhibited in Fig. 5.

Table 4 AIC results to check the distributions of the positive rainfall datasets.

Rainfall station	Densities	
	Normal	Lognormal	Cauchy	gamma	
Irrigation Office I	214.35	202.35	218.39	197.47	
Mae Taeng Project (January)	214.66	198.63	225.36	195.98	
Mae Taeng Project (February)	144.76	140.28	153.50	135.93	
Mae Hong Huk	216.84	205.73	223.86	204.57	
Mae Kuang	701.53	629.15	691.32	612.84	

Figure 5 Q–Q plots for distribution fitting of the positive rainfall data from Irrigation Office Station I, Chiang Mai city.

The zero values in rainfall data fitted a binomial distribution, and so the delta-gamma distribution is suitable for this data. The summary statistics were computed for the rainfall dataset from Irrigation Office Station I as n = 50, n(0) = 27, n(1) = 23 with maximum likelihood estimators δ^ = 0.54, α^ = 5.30, β^ = 2.06, and τ^ = 1.18. The 95% confidence intervals for τ are reported in Table 5. In accordance with the simulation results in the previous section, the length of the Bayesian HPD interval based on the Jeffrey’s rule prior was shorter than the other methods, thereby confirming its suitability for constructing the confidence interval for the mean of a delta-gamma distribution.

Table 5 The 95% confidence intervals for the mean for the rainfall dataset from Irrigation Office Station I, Chiang Mai city.

Methods	Confidence intervals for τ	Length of intervals	
	Lower bound	Upper bound		
Bayesian: The Jeffrey’s rule (Credible)	9.515	18.583	9.068	
Bayesian: The Jeffrey’s rule (HPD)	9.295	18.229	8.934	
Bayesian: The uniform (Credible)	10.185	21.117	10.932	
Bayesian: The uniform (HPD)	9.469	20.059	10.590	
Fiducial quantities (FQ)	7.248	19.671	12.423	

Example 2: testing the difference between the means of two delta-gamma distributions with equal sample sizes

Since January and February are in the winter season, they have similar precipitation profiles containing both positive and zero observations, and so the data were found to be consistent with a delta-gamma distribution. Therefore, the data from these months were chosen to compare the difference between the means of two delta-gamma distributions in this study. For n = m, we used monthly rainfall data from the Mae Taeng Project station, Mae Taeng district, Chiang Mai province. There were 46 observations for comparing rainfall data from the same station in January and February from 1976 to 2021. The densities of the rainfall data are shown in Fig. 6.

Figure 6 The densities of the rainfall data from Mae Taeng Project station, Mae Taeng district, Chiang Mai province, for (A) January and (B) February from 1976 to 2021.

Next, we tested the distributions of the positive rainfall datasets from the Mae Taeng Project station using AIC, the results of which are reported in Table 4. Q–Q plots of positive rainfall data showing that they follow gamma distributions are exhibited in Fig. 7.

Figure 7 Q–Q plots for distribution fitting of the positive rainfall data from the Mae Taeng Project station, Mae Taeng district, Chiang Mai province, for (A) January and (B) February from 1976 to 2021.

The summary statistics were computed for the rainfall in January dataset from the Mae Taeng Project station as n = 46, n(0) = 23, n(1) = 23, δ^ = 0.50, α^ = 4.41, β^ = 1.77, and τ^ = 1.25 and for the rainfall in February dataset as m = 46, m(0) = 29, m(1) = 17, δ^2 = 0.63, α^2 = 4.96, β^2 = 2.15, and τ^2 = 0.85. From the 95% confidence intervals for ψ (Table 6), the expected length of the Bayesian HPD interval based on Jeffrey’s rule prior was shorter than the other methods, which confirmed its suitability for constructing confidence intervals for the difference between the means of delta-gamma distributions with equal sample sizes.

Table 6 95% confidence intervals for the difference between the means of the rainfall datasets from the Mae Taeng Project station, Mae Taeng district, Chiang Mai province.

Methods	Confidence intervals for ψ	Length of intervals	
	Lower bound	Upper bound		
Bayesian: The Jeffrey’s rule (Credible)	−5.709	7.533	13.242	
Bayesian: The Jeffrey’s rule (HPD)	−6.025	7.105	13.130	
Bayesian: The uniform (Credible)	−7.364	8.726	16.090	
Bayesian: The uniform (HPD)	−7.296	8.776	16.072	
Fiducial quantities (FQ)	−2.155	15.764	17.919	

Example 3: testing the difference between the means of two delta-gamma distributions with unequal sample sizes

For n ≠ m, we used monthly rainfall data from the Mae Hong Huk station, Doi Saket district, Chiang Mai province (51 observations for January and March 2005–2021) and the Mae Kuang station, Doi Saket district, Chiang Mai province (171 observations for January and March 1965–2021) to compare the rainfall data from two stations in the same district for the same months. The densities of the rainfall data are shown in Fig. 8.

Figure 8 The densities of the rainfall data from the (A) Mae Hong Huk and (B) Mae Kuang station, Doi Saket district, Chiang Mai province.

The results in Table 4 are from using AIC to test the suitability of distributions to fit the positive rainfall datasets for the two stations, while the Q–Q plots of the positive rainfall data in Fig. 9 show that they follow gamma distributions.

Figure 9 Q–Q plots for distribution fitting of the rainfall data from the (A) Mae Hong Huk and (B) Mae Kuang stations, Doi Saket district, Chiang Mai province.

The summary statistics for the rainfall dataset were n = 51, n(0) = 28, n(1) = 23, δ^ = 0.55, α^ = 9.01, β^ = 3.20, and τ^ = 1.27 for the Mae Hong Huk station and m = 171, m(0) = 97, m(1) = 74, δ^2 = 0.57, α^2 = 3.91, β^2 = 1.60, and τ^2 = 1.06 for the Mae Kuang station. The 95% confidence intervals for ψ reported in Table 7 show that the length of the Bayesian HPD interval based on Jeffrey’s rule prior was shorter than the others, which confirmed its appropriateness for constructing confidence intervals for the difference between the means of delta-gamma distributions with unequal sample sizes.

Table 7 95% confidence intervals for the difference between the means of the rainfall datasets from the Mae Taeng Project station, Mae Taeng district, Chiang Mai province.

Methods	Confidence intervals for ψ	Length of intervals	
	Lower bound	Upper bound		
Bayesian: Jeffrey’s rule (Credible)	−2.530	8.334	10.864	
Bayesian: Jeffrey’s rule (HPD)	−2.561	8.272	10.833	
Bayesian: The uniform (Credible)	−2.560	9.503	12.063	
Bayesian: The uniform (HPD)	−2.619	9.357	11.976	
Fiducial quantities (FQ)	−2.832	10.864	13.696	

Discussion

We used Krishnmoorthy & Wang’s (2016) approach for establishing confidence intervals for the mean of a gamma distribution by using FQs in the case of the same distribution with excess zeros. Furthermore, we extended Yosboonruang, Niwitpong & Niwitpong’s (2019) approach for building confidence intervals for distributions containing some zero observations by using Bayesian methods based on Jeffrey’s rule and uniform priors. The results show that the Bayesian HPD interval with Jeffrey’s rule prior performed well in terms of coverage probability and had the shortest expected lengths for estimating the mean and the difference between the means of delta-gamma distributions with equal sample sizes. For unequal sample sizes, the results of simulation for the difference between the means of two delta-gamma distributions for n > m were similar to n < m. The proposed strategy can be utilized to help mitigate droughts or floods caused by insufficient or excessive rainfall, respectively. Similarly, the government could use our approach to control the output from dams when there is insufficient or too much rain. However, a limitation of the study is that we cannot apply our method to real-world data that does not contain zero observations, such as often occurs in rainfall data observations during the rainy season.

Conclusions

We constructed confidence intervals for the mean and difference between the means of two delta-gamma distributions using FQs and Bayesian methods based on the Jeffrey’s rule and uniform priors. The performances of the confidence intervals were evaluated in terms of their coverage probabilities and expected lengths. The results of a simulation study show that the coverage probabilities of the Bayesian HPD interval based on the Jeffrey’s rule prior were greater than or close to the nominal confidence level of 0.95 in almost all cases and its expected length was shorter than the other methods for both the mean and the difference between the means of two delta-gamma distributions. When using rainfall datasets to illustrate the efficacies of the proposed methods using real data, the Bayesian HPD interval based on the Jeffrey’s rule prior performed better than the other methods in terms of interval length, which is consistent with the simulation results. Therefore, the Bayesian HPD interval based on the Jeffrey’s rule prior is recommended for constructing confidence intervals for the mean and the difference between the means of two delta-gamma distributions.

Supplemental Information

Supplemental Information 1 Dataset 1: Monthly rainfall data (mm) from Irrigation Office I Station in January, Chiang Mai city.

Irrigation Office I Station in January, Chiang Mai city, comprising 50 observations from 1972 to 2021.

Click here for additional data file.

Supplemental Information 2 Dataset 2: Monthly rainfall data (mm) from Mae Taeng Project Station in January, Mae Taeng district, Chiang Mai province.

There were 46 observations from 1976 to 2021.

Click here for additional data file.

Supplemental Information 3 Dataset 3: Monthly rainfall data (mm) from Mae Taeng Project Station in February, Mae Taeng district, Chiang Mai province.

There were 46 observations from 1976 to 2021.

Click here for additional data file.

Supplemental Information 4 Dataset 4: Monthly rainfall data (mm) from Mae Hong Huk Station, Doi Saket district, Chiang Mai province.

There were 51 observations for January and March 2005 to 2021.

Click here for additional data file.

Supplemental Information 5 Dataset 5: Monthly rainfall data (mm) from Mae Kuang Station, Doi Saket district, Chiang Mai province.

There were 171 observations for January and March 1965 to 2021.

Click here for additional data file.

Supplemental Information 6 R code 1 to compute the coverage probability and average length width of all confidence interval for single mean of DG distribution.

Click here for additional data file.

Supplemental Information 7 R code 2 to compute the coverage probability and average length width of all confidence interval for the difference between means of DG distribution.

Click here for additional data file.

Supplemental Information 8 R code 3 to apply with rainfall data For the Single Mean of Delta-gamma Distributions: Table 5.

Click here for additional data file.

Supplemental Information 9 R code 4 to apply with rainfall data For the Single Mean of Delta-gamma Distributions: Table 6.

Click here for additional data file.

Supplemental Information 10 R code 5 to apply with rainfall data For the Single Mean of Delta-gamma Distributions: Table 7.

Click here for additional data file.

Supplemental Information 11 R code to estimate AIC results.

Click here for additional data file.

Additional Information and Declarations

Competing Interests

Author Contributions

Data Availability

The authors declare that they have no competing interests.

Theerapong Kaewprasert conceived and designed the experiments, performed the experiments, analyzed the data, prepared figures and/or tables, authored or reviewed drafts of the paper, and approved the final draft.

Sa-Aat Niwitpong performed the experiments, authored or reviewed drafts of the paper, and approved the final draft.

Suparat Niwitpong conceived and designed the experiments, performed the experiments, analyzed the data, prepared figures and/or tables, authored or reviewed drafts of the paper, and approved the final draft.

The following information was supplied regarding data availability:

The raw data are available in the Supplemental Files.

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
