# Peer review of "Bayesian estimation for the mean of delta-gamma distributions with application to rainfall data in Thailand"

_PeerJ, doi:10.7717/peerj.13465_

## Round 0.1 · original submission · Minor Revisions

The reviewers agree the work is solid and the results are reasonable. There are suggested changes from the three reviewers before the article is published. Please revise your manuscript accordingly.

·

Basic reporting

In this manuscript, credible interval (CI) and highest posterior density (HPD) interval for the mean and the difference between the means of the delta-gamma distribution are constructed. For Bayesian method, the Jeffreys’ prior and uniform prior are taken into consideration. Also the CI and HPD intervals are derived based on fiducial quantities. The numerical computation is performed for comparing the efficacy of five proposed methods in terms of their coverage probabilities and expected lengths.

Experimental design

The structure of the manuscript is satisfactory. The paper is logically and technically in standard paper format. The computation part and simulation is reasonable.

Validity of the findings

The results and Tables are clear and acceptable. In general, after some minor revisions, the paper is suitable for acceptance.

Additional comments

I invite the authors to make minor revisions to improve the presentation, the comments are listed below:

1. Motivation of the paper needs to be illustrated.

2. In equation (8), the last term of the joint posterior density of θ will be exp(- x_n(1) /(2σ^2 )) instead of exp( x_n(1) /(2σ^2 )). Also the equations (10), (20) and (22) need to be corrected in similar manner.

3. Each equation should be ended either with “.” Or with “,”.

4. The resolution of all the plots (Figures 1 - 9) needs to be improved. I suggest to produce your graphics in PDF (do not convert it from png, jpeg, … , to pdf).

5. The line before equation (17), “Thus, the Fisher information …”, after the word ‘for’, "ϕ” is missing.

Reviewer 2 ·

Basic reporting

The authors presented well the methods and provided examples. They also uploaded the data files and the R code.

I have minor comments: In the abstract, discuss in one to two sentences at the start of the paragraph why this research is needed (or state the research question). The methods are acceptable, but I suggest checking the manuscript for typographical errors (especially regarding the English style).

Experimental design

The paper used data from weather stations. The authors uploaded the data files as supplementary information, which is good and highly encouraged. However, I recommend including in the paper a statement that the authors asked permission from the stations to publish and use the data for this research.

Validity of the findings

The methods are acceptable. Results can also be replicated as the authors uploaded the R files, and they provided examples.

Reviewer 3 ·

Basic reporting

1. In the introduction, rainfall is important enough. The review of rainfall is need more.

Experimental design

1. The independent of random sample or random variable is need to declare. It effects to your derivation of the likelihood function.
2. The description of HPD interval is need to explain more in both lines 89-92 and Algorithm 1.

Validity of the findings

1. In the results of simulation for the difference between the means of two delta-gamma distribution, the case of n>m is need to show or explain. Is it similar or difference from n<m.
2. Rainfall data from Example 1, were obtained by monthly with 50 observations from 1972-2021. Is it monthly?
3. Why is reason of data collection of Example 2 by January and February from 1976-2021.
4. The period of observations collections of 3 examples in the part of an empirical application is need to explain of reason in the manuscript.

Additional comments

1. The discussion should be separated from the results.
2. Add the limitation of your study in the discussion.

---

## Round 0.2 · accepted · Accept

The revision is sufficient to address the comments from the reviewers. The manuscript is clear and easy to follow. It can serve as a document for governments and researchers, who are interested in applying the delta-gamma distribution to rainfall data.

---

## Author Rebuttal · Round 0.2

# Response to Reviewers and Editor

**Journal:**PeerJ
**Manuscript ID:** 70891
**Title:** Bayesian estimation for the mean of delta-gamma distributions with application to rainfall data in Thailand
**Authors:** Theerapong Kaewprasert, Sa-Aat Niwitpong, and Suparat Niwitpong

Dear Reviewers and Editor,
We are grateful for the reviewer's valuable comments and have all suggestions seriously. Reviewer's critiques addressed section by section in this document, and corrections were incorporated in manuscript accordingly.

*Editor comments (Lei Wang)*

*MINOR REVISIONS*

*The reviewers agree the work is solid and the results are reasonable. There are suggested changes from the three reviewers before the article is published. Please revise your manuscript accordingly.*

*Reviewer 1 (Proloy Banerjee)*

*Basic reporting*
*No comment*

*Experimental design*
*No comment*

*Validity of the findings*
*No comment*

*Additional comments*
*I invite the authors to make minor revisions to improve the presentation, the comments are listed below:*

*1. Motivation of the paper needs to be illustrated.*

**Response:** In the introduction, the authors review the rainfall at the beginning of the paragraph.
"Weather conditions can vary immensely each day and forecasting it accurately for up to 7 days in advance is greatly desired. The climate in a given area provides a broad picture of temperature and rainfall variation over time and is categorized into seasons. For example, Chiang Mai, a province in Northern Thailand, has three seasons: summer (from March to June), the rainy season (from July to October), and winter (from November to February). The major economic output from Chiang Mai is from agriculture, for which rainfall is essential: insufficient or nonexistent rainfall (drought conditions)

causes crops to die whereas excessive rainfall (flooding) destroys crops and can cause disasters such as landslides. Therefore, predicting the amount of rainfall during each period is very important because it would enable farmers to plan the proper use of water resources accordingly. Thus, assessing rainfall dispersion in specific areas by using statistical methods such as the mean is of great importance. Chiang Mai has an average rainfall of approximately 1,134 mm per year, with the highest rainfall in a day being166.5 mm (August 14, 1968). The rainiest month is August and the least rainy month is January (Amatayakul and Chomtha, 2013). There can be zero millimeters of rainfall in a month, and so a monthly rainfall series often includes zero values. When a rainwater series only contains positive values, they can be fitted to standard continuous probability distributions such as a gamma distribution. For instance, Sangnawakij and Niwitpong (2017) and Krishnmoorthy et al. (2008) constructed confidence interval for a gamma distribution of monthly rainfall data. However, a delta-gamma distribution (or a zero-inflated gamma distribution) is more suitable for data containing both positive and zero observations. The positive values comprise a gamma distribution with shape and rate parameters while the zero values follow a binomial distribution with proportion of zeros. Inference from a delta-gamma distribution applied to real data has been conducted in many fields. For instance, the testing of body armor for stab resistance in engineering during which a zero value was recorded when the arm or was not pierced (Zimmer et al., 2020) and ecological data for biomasses that often contain a high proportion of zeros with skewed positive values (Lecomte et al., 2013)."

*2. In equation (8), the last term of the joint posterior density of $\vartheta$ will be exp(- x_n(1) /(2σ^2 )) instead of exp( x_n(1) /(2σ^2 )). Also the equations (10), (20) and (22) need to be corrected in similar manner.*

**Response:** In equations (8), (10), (20) and (22), the authors revised the last term of the joint posterior density of θ to be exp(- x_n(1) /(2σ^2 )) instead of exp( x_n(1) /(2σ^2 )).

In equation (8),

$$p(\theta|x) = \frac{1}{Beta\left(n_{(0)} + \frac{1}{2}, n_{(1)} + \frac{3}{2}\right)} \delta^{n_{(0)} - \frac{1}{2}} (1-\delta)^{n_{(1)} + \frac{1}{2}}$$

$$\times \frac{\sqrt{n_{(1)}}}{\sqrt{2\pi\sigma^2}} \exp\left(-\frac{n_{(1)}}{2\sigma^2}(\mu - \hat{\mu})^2\right) \times \frac{\left(\frac{x_{n_{(1)}}}{2}\right)^{\frac{n_{(1)}}{2}}}{\Gamma\left(\frac{n_{(1)}}{2}\right)} \left(\sigma^2\right)^{-1-\frac{n_{(1)}}{2}} \exp\left(-\frac{x_{n_{(1)}}}{2\sigma^2}\right), \qquad (8)$$

In equation (10),

$$p(\theta|x) = \frac{1}{Beta\left(n_{(0)} + 1, n_{(1)} + 1\right)} \delta^{n_{(0)}} (1-\delta)^{n_{(1)}}$$

$$\times \frac{\sqrt{n_{(1)}}}{\sqrt{2\pi\sigma^2}} \exp\left(-\frac{n_{(1)}}{2\sigma^2}(\mu - \hat{\mu})^2\right) \times \frac{\left(\frac{x_{n_{(1)}}}{2}\right)^{\frac{n_{(1)}-3}{2}}}{\Gamma\left(\frac{n_{(1)}-3}{2}\right)} \left(\sigma^2\right)^{-1-\frac{(n_{(1)}-3)}{2}} \exp\left(-\frac{x_{n_{(1)}}}{2\sigma^2}\right), \qquad (10)$$

In equation (20),

$$p(\phi|x,v) = p(\theta|x) \times \frac{1}{Beta\left(m_{(0)} + \frac{1}{2}, m_{(1)} + \frac{3}{2}\right)} \delta_2^{m_{(0)} - \frac{1}{2}} (1 - \delta_2)^{m_{(1)} + \frac{1}{2}}$$

$$\times \frac{\sqrt{m_{(1)}}}{\sqrt{2\pi\sigma_2^2}} \exp\left(-\frac{m_{(1)}}{2\sigma_2^2}(\mu_2 - \hat{\mu}_2)^2\right)$$

$$\times \frac{\left(\frac{v_{m_{(1)}}}{2}\right)^{\frac{m_{(1)}}{2}}}{\Gamma\left(\frac{m_{(1)}}{2}\right)} (\sigma_2^2)^{-1 - \frac{m_{(1)}}{2}} \exp\left(-\frac{v_{m_{(1)}}}{2\sigma_2^2}\right), \tag{20}$$

In equation (22),

$$p(\phi|x,v) = p(\theta|x) \times \frac{1}{Beta\left(m_{(0)} + 1, m_{(1)} + 1\right)} \delta_2^{m_{(0)}} (1 - \delta_2)^{m_{(1)}}$$

$$\times \frac{\sqrt{m_{(1)}}}{\sqrt{2\pi\sigma_2^2}} \exp\left(-\frac{m_{(1)}}{2\sigma_2^2}(\mu_2 - \hat{\mu}_2)^2\right)$$

$$\times \frac{\left[\frac{v_{m_{(1)}}}{2}\right]^{\frac{m_{(1)} - 3}{2}}}{\Gamma\left(\frac{m_{(1)} - 3}{2}\right)} (\sigma_2^2)^{-1 - \frac{(m_{(1)} - 3)}{2}} \exp\left(-\frac{v_{m_{(1)}}}{2\sigma_2^2}\right). \tag{22}$$

*3. Each equation should be ended either with "." or with ",".*

**Response:** For each equation, the authors were reviewed and improved, and the ending should be either "." or ",".

*4. The resolution of all the plots (Figures 1 - 9) needs to be improved. I suggest to produce your graphics in PDF (do not convert it from png, jpeg, … , to pdf).*

**Response:** All of the plots were resolved (Figures 1 - 9), the resolution of the figures has been increased to meet the criteria of the journal.

*5. The line before equation (17), "Thus, the Fisher information …", after the word 'for', "φ" is missing.*

**Response:** After the word "for" in the sentence, the authors *add* "φ".

*Reviewer 2*

**Basic reporting**
*I have minor comments: In the abstract, discuss in one to two sentences at the start of the paragraph why this research is needed (or state the research question). The methods are acceptable, but I suggest checking the manuscript for typographical errors (especially regarding the English style).*

**Response:** Thank you for your suggestion. The authors were re-evaluated in order to double-check the manuscript for typographical errors (especially regarding the English style). In the abstract, the authors discuss in one to two sentences at the start of the paragraph why this research is needed (or state the research question).

"Precipitation and flood forecasting are difficult due to rainfall variability. The mean of a delta-gamma distribution can be used to analyze rainfall data for predicting future rainfall, thereby reducing the risks of future disasters due to excessive or too little rainfall. In this study, we construct credible and highest posterior density (HPD) intervals for the mean and the difference between the means of delta-gamma distributions by using Bayesian methods based on Jeffreys'rule and uniform priors along with a confidence interval based on fiducial quantities. The results of a simulation study indicate that the Bayesian HPD interval based on Jeffreys'rule prior performed well in terms of coverage probability and provided the shortest expected length. Rainfall data from Chiang Mai province, Thailand, are also used to illustrate the efficacies of the proposed methods."

**Experimental design**
*The paper used data from weather stations. The authors uploaded the data files as supplementary information, which is good and highly encouraged. However, I recommend including in the paper a statement that the authors asked permission from the stations to publish and use the data for this research.*

**Response:** Thank you for your suggestion. The Upper Northern Region Irrigation Hydrology Center (http://hydro-1.rid.go.th/) provided free monthly rainfall data from Chiang Mai province, Thailand, which was used in this study. As a result, these data can be freely exploited.

Line 233, the authors have included a source for the rainfall data before the word "using the five…".

**Validity of the findings**
*No comment*

*Reviewer 3*

*Basic reporting*
1. In the introduction, rainfall is important enough. The review of rainfall is need more.

**Response:** In the introduction, the authors review the rainfall at the beginning of the paragraph.
"Weather conditions can vary immensely each day and forecasting it accurately for up to 7 days in advance is greatly desired. The climate in a given area provides a broad picture of temperature and rainfall variation over time and is categorized into seasons. For example, Chiang Mai, a province in Northern Thailand, has three seasons: summer (from March to June), the rainy season (from July to October), and winter (from November to February). The major economic output from Chiang Mai is from agriculture, for which rainfall is essential: insufficient or nonexistent rainfall (drought conditions) causes crops to die whereas excessive rainfall (flooding) destroys crops and can cause disasters such as landslides. Therefore, predicting the amount of rainfall during each period is very important because it would enable farmers to plan the proper use of water resources accordingly. Thus, assessing rainfall dispersion in specific areas by using statistical methods such as the mean is of great importance. Chiang Mai has an average rainfall of approximately 1,134 mm per year, with the highest rainfall in a day being166.5 mm (August 14, 1968). The rainiest month is August and the least rainy month is January (Amatayakul and Chomtha, 2013). There can be zero millimeters of rainfall in a month, and so a monthly rainfall series often includes zero values. When a rainwater series only contains positive values, they can be fitted to standard continuous probability distributions such as a gamma distribution. For instance, Sangnawakij and Niwitpong (2017) and Krishnmoorthy et al. (2008) constructed confidence interval for a gamma distribution of monthly rainfall data. However, a delta-gamma distribution (or a zero-inflated gamma distribution) is more suitable for data containing both positive and zero observations. The positive values comprise a gamma distribution with shape and rate parameters while the zero values follow a binomial distribution with proportion of zeros. Inference from a delta-gamma distribution applied to real data has been conducted in many fields. For instance, the testing of body arm or for stab resistance in engineering during which a zero value was recorded when the arm or was not pierced (Zimmer et al., 2020) and ecological data for biomasses that often contain a high proportion of zeros with skewed positive values (Lecomte et al., 2013)."

*Experimental design*
1. The independent of random sample or random variable is need to declare. It effects to your derivation of the likelihood function.

**Response:** 1) The line after line 71, the authors were reviewed and improved as follows:
"Let $X = (X_1, X_2, ..., X_n)$ be independent and identically distributed random sample from a delta-gamma distribution denoted as $X \sim \mathrm{DG}(\alpha, \beta, \delta)$."

2) Two lines after line 75, the authors have added the following statements between the words "distribution" and "is approximately...".
"Suppose that $G = (G_1, G_2, ..., G_n)$ be independent and identically distributed random variables from a gamma distribution, denoted as $G(\alpha, \beta)$, and that $Y = G^{1/3} \sim \mathrm{N}(\mu, \sigma^2)$."

3) The two lines before equation (5) and after the word "$Y \sim \mathrm{N}(\mu, \sigma^2)$". The authors describe variable $Y$ as follows:

"Suppose that $Y = (Y_1, Y_2, ..., Y_n)$ be independent and identically distributed random variables with probability density function, the likelihood function is $p(y|\lambda) \propto (\sigma^2)^{-n_{(1)}/2} \exp\left(-\frac{1}{2\sigma^2}\sum_{i=1}^{n_{(1)}}(y_i - \mu)^2\right)$"

4) The line 118, the authors were reviewed and improved as follows:
"Suppose that $X = (X_1, X_2, ..., X_n)$ be independent and identically distributed random variables"

5) The line before equation (15), the authors were reviewed and improved as follows:
"Let $X = (X_1, X_2, ..., X_n)$ and $V = (V_1, V_2, ..., V_m)$ be independent and identically distributed random samples from two delta-gamma distributions denoted as $X \sim \mathrm{DG}(\alpha, \beta, \delta)$ and $V \sim \mathrm{DG}(\alpha_2, \beta_2, \delta_2)$, then the difference between their means is simply"

*2. The description of HPD interval is need to explain more in both lines 89-92 and Algorithm 1.*

**Response:** The HPD interval, as defined by Box and Tiao (1973), is described in lines 78-83 by the authors.

After the word "...R software suite," the authors have inserted "defined by Box and Tiao (1973)" on line 92.

The authors changed and improved Algorithm 1 as follows:
- Step 6 compute the 95% Bayesian credible, as given in Eq. (9) from $CI_J$ and HPD interval of $\tau_J$ from $CI_{\mathrm{HPD.J}}$, defined by Box and Tiao (1973);

***Validity of the findings***
*1. In the results of simulation for the difference between the means of two delta-gamma distribution, the case of n>m is need to show or explain. Is it similar or difference from n<m.*

**Response:** It is similar to cases n<m. From line 220 onwards, the author proceeds to explain.

"Furthermore, the results for the difference between the means of two delta-gamma distributions for sample size n>m yielded similar results to those for n<m."

For n>m, the simulation results are similar to cases n<m. As can be seen from table 1 below.

Table 1. Coverage probabilities and expected lengths of the 95% confidence intervals for the difference between the means of two delta-gamma distributions (n>m).

| (n, m) | (δ, δ₂) | (α, β) | (α₂, β₂) | Coverage probability (Expected length) | | | | |
|---|---|---|---|---|---|---|---|---|
| | | | | $CI_{d.J}$ | $CI_{d.HPD.J}$ | $CI_{d.U}$ | $CI_{d.HPD.U}$ | $CI_{d.FO}$ |
| (50, 30) | (0.2, 0.2) | (5.5, 2.0) | (5.5, 2.0) | 0.9667 (0.9966) | 0.9626 (0.9918) | 0.9724 (1.0357) | 0.9708 (1.0309) | 0.9865 (1.2205) |
| | | (6.0, 2.0) | (5.5, 1.0) | 0.9552 (1.6904) | 0.9559 (1.6758) | 0.9617 (1.7665) | 0.9634 (1.7538) | 0.9854 (2.0826) |
| | | (5.5, 2.0) | (6.0, 1.0) | 0.9588 (1.7930) | 0.9596 (1.7749) | 0.9629 (1.8717) | 0.9664 (1.8561) | 0.9860 (2.1737) |
| | | (6.0, 1.0) | (6.0, 1.0) | 0.9714 (2.1594) | 0.9679 (2.1488) | 0.9782 (2.2402) | 0.9754 (2.2298) | 0.9888 (2.5813) |
| | (0.5, 0.5) | (2.0, 2.0) | (2.0, 2.0) | 0.9581 | 0.9598 | 0.9705 | 0.9750 | 0.9849 |

| (n, m) | | | | Col1 | Col2 | Col3 | Col4 | Col5 |
|---|---|---|---|---|---|---|---|---|
| | | | | (0.4865) | (0.4839) | (0.5347) | (0.5301) | (0.6177) |
| | | (2.5, 2.0) | (2.0, 1.0) | 0.9551 | 0.9553 | 0.9696 | 0.9721 | 0.9830 |
| | | | | (0.8438) | (0.8383) | (0.9375) | (0.9250) | (1.0689) |
| | | (2.0, 2.0) | (2.5, 1.0) | 0.9660 | 0.9655 | 0.9765 | 0.9763 | 0.9848 |
| | | | | (0.9675) | (0.9619) | (1.0569) | (1.0455) | (1.1777) |
| | | (2.5, 1.0) | (2.5, 1.0) | 0.9723 | 0.9731 | 0.9798 | 0.9813 | 0.9868 |
| | | | | (1.1715) | (1.1661) | (1.2608) | (1.2525) | (1.4125) |
| | (0.7, 0.7) | (1.25, 2.0) | (1.25, 2.0) | 0.9494 | 0.9632 | 0.9706 | 0.9881 | 0.9798 |
| | | | | (0.3288) | (0.3237) | (0.4807) | (0.4521) | (0.4453) |
| | | (1.5, 2.0) | (1.25, 1.0) | 0.9471 | 0.9543 | 0.9715 | 0.9840 | 0.9786 |
| | | | | (0.5639) | (0.5486) | (0.8614) | (0.7808) | (0.7705) |
| | | (1.25, 2.0) | (1.5, 1.0) | 0.9574 | 0.9607 | 0.9784 | 0.9881 | 0.9842 |
| | | | | (0.6177) | (0.6020) | (0.9076) | (0.8279) | (0.8189) |
| | | (1.5, 1.0) | (1.5, 1.0) | 0.9577 | 0.9678 | 0.9780 | 0.9883 | 0.9830 |
| | | | | (0.7407) | (0.7315) | (1.0345) | (0.9812) | (0.9769) |
| (100, 50) | (0.2, 0.2) | (5.5, 2.0) | (5.5, 2.0) | 0.9623 | 0.9608 | 0.9650 | 0.9641 | 0.9845 |
| | | | | (0.7501) | (0.7464) | (0.7665) | (0.7628) | (0.9002) |
| | | (6.0, 2.0) | (5.5, 1.0) | 0.9586 | 0.9600 | 0.9606 | 0.9620 | 0.9858 |
| | | | | (1.3050) | (1.2948) | (1.3379) | (1.3283) | (1.5731) |
| | | (5.5, 2.0) | (6.0, 1.0) | 0.9624 | 0.9640 | 0.9637 | 0.9657 | 0.9869 |
| | | | | (1.3898) | (1.3776) | (1.4245) | (1.4131) | (1.6453) |
| | | (6.0, 1.0) | (6.0, 1.0) | 0.9692 | 0.9670 | 0.9719 | 0.9712 | 0.9866 |
| | | | | (1.6288) | (1.6206) | (1.6634) | (1.6553) | (1.9087) |
| | (0.5, 0.5) | (2.0, 2.0) | (2.0, 2.0) | 0.9572 | 0.9568 | 0.9639 | 0.9644 | 0.9825 |
| | | | | (0.3594) | (0.3578) | (0.3740) | (0.3721) | (0.4389) |
| | | (2.5, 2.0) | (2.0, 1.0) | 0.9615 | 0.9608 | 0.9691 | 0.9681 | 0.9858 |
| | | | | (0.6362) | (0.6331) | (0.6653) | (0.6612) | (0.7744) |
| | | (2.0, 2.0) | (2.5, 1.0) | 0.9713 | 0.9706 | 0.9769 | 0.9750 | 0.9867 |
| | | | | (0.7418) | (0.7384) | (0.7696) | (0.7654) | (0.8664) |
| | | (2.5, 1.0) | (2.5, 1.0) | 0.9754 | 0.9744 | 0.9785 | 0.9785 | 0.9873 |
| | | | | (0.8740) | (0.8703) | (0.9013) | (0.8972) | (1.0140) |
| | (0.7, 0.7) | (1.25, 2.0) | (1.25, 2.0) | 0.9464 | 0.9539 | 0.9591 | 0.9695 | 0.9783 |
| | | | | (0.2275) | (0.2255) | (0.2580) | (0.2535) | (0.2872) |
| | | (1.5, 2.0) | (1.25, 1.0) | 0.9502 | 0.9544 | 0.9632 | 0.9698 | 0.9797 |
| | | | | (0.4001) | (0.3942) | (0.4609) | (0.4479) | (0.5063) |
| | | (1.25, 2.0) | (1.5, 1.0) | 0.9594 | 0.9605 | 0.9702 | 0.9747 | 0.9824 |
| | | | | (0.4469) | (0.4406) | (0.5060) | (0.4931) | (0.5491) |
| | | (1.5, 1.0) | (1.5, 1.0) | 0.9630 | 0.9670 | 0.9723 | 0.9776 | 0.9828 |
| | | | | (0.5225) | (0.5186) | (0.5805) | (0.5725) | (0.6396) |
| (200, 100) | (0.2, 0.2) | (5.5, 2.0) | (5.5, 2.0) | 0.9625 | 0.9609 | 0.9646 | 0.9625 | 0.9851 |
| | | | | (0.5337) | (0.5312) | (0.5394) | (0.5369) | (0.6321) |
| | | (6.0, 2.0) | (5.5, 1.0) | 0.9604 | 0.9611 | 0.9617 | 0.9624 | 0.9861 |
| | | | | (0.9305) | (0.9247) | (0.9419) | (0.9363) | (1.1027) |
| | | (5.5, 2.0) | (6.0, 1.0) | 0.9648 | 0.9645 | 0.9648 | 0.9658 | 0.9859 |
| | | | | (0.9924) | (0.9858) | (1.0046) | (0.9980) | (1.1541) |
| | | (6.0, 1.0) | (6.0, 1.0) | 0.9710 | 0.9692 | 0.9721 | 0.9706 | 0.9878 |
| | | | | (1.1598) | (1.1544) | (1.1719) | (1.1665) | (1.3400) |
| | (0.5, 0.5) | (2.0, 2.0) | (2.0, 2.0) | 0.9603 | 0.9604 | 0.9632 | 0.9624 | 0.9834 |
| | | | | (0.2521) | (0.2510) | (0.2562) | (0.2551) | (0.3004) |
| | | (2.5, 2.0) | (2.0, 1.0) | 0.9614 | 0.9595 | 0.9649 | 0.9642 | 0.9842 |
| | | | | (0.4479) | (0.4460) | (0.4558) | (0.4538) | (0.5307) |
| | | (2.0, 2.0) | (2.5, 1.0) | 0.9714 | 0.9707 | 0.9728 | 0.9722 | 0.9853 |
| | | | | (0.5240) | (0.5218) | (0.5317) | (0.5294) | (0.5963) |
| | | (2.5, 1.0) | (2.5, 1.0) | 0.9756 | 0.9750 | 0.9771 | 0.9768 | 0.9869 |
| | | | | (0.6172) | (0.6147) | (0.6248) | (0.6222) | (0.6975) |
| | (0.7, 0.7) | (1.25, 2.0) | (1.25, 2.0) | 0.9507 | 0.9526 | 0.9564 | 0.9584 | 0.9788 |
| | | | | (0.1537) | (0.1528) | (0.1603) | (0.1592) | (0.1868) |
| | | (1.5, 2.0) | (1.25, 1.0) | 0.9508 | 0.9515 | 0.9574 | 0.9594 | 0.9796 |
| | | | | (0.2710) | (0.2688) | (0.2841) | (0.2813) | (0.3286) |

| | | (1.25, 2.0) | (1.5, 1.0) | 0.9624 (0.3056) | 0.9634 (0.3031) | 0.9672 (0.3182) | 0.9672 (0.3152) | 0.9836 (0.3597) |
| | | (1.5, 1.0) | (1.5, 1.0) | 0.9640 (0.3582) | 0.9663 (0.3563) | 0.9684 (0.3707) | 0.9703 (0.3686) | 0.9840 (0.4202) |

*2. Rainfall data from Example 1, were obtained by monthly with 50 observations from 1972-2021. Is it monthly?*

**Response:** It is monthly. The authors add "in January" to the sentence on lines 235-237.

"We used monthly rainfall data (mm) from Irrigation Office Station I, Chiang Mai city, comprising 50 observations in January from 1972–2021."

*3. Why is reason of data collection of Example 2 by January and February from 1976-2021.*

**Response:** Next to line 250, the authors describe the reason for the data collection in Example 2.

"Since January and February are in the winter season, they have similar precipitation profiles containing both positive and zero observations, and so the data were found to be consistent with a delta-gamma distribution. Therefore, the data from these months were chosen to compare the difference between the means of two delta-gamma distributions in this study."

*4. The period of observations collections of 3 examples in the part of an empirical application is need to explain of reason in the manuscript.*

**Response:** After line 233, the authors describe the period of observations collections of three examples.

"There are three cases as follows. Case 1 was used to test the mean of a delta-gamma distribution for which we used rainfall data from only one rain station in Chiang Mai to provide a sample size consistent with that used in the simulation study. The difference between the means of two delta-gamma distributions with equal sample sizes was investigated in Case 2 by using rainfall data over a period of time at the same station in Chiang Mai for various months within the same season. For Case 3, we compared the means of two delta-gamma distributions with uneven sample sizes by combining data from several stations in Chiang Mai for the same month."

***Additional comments***
*1. The discussion should be separated from the results.*

**Response:** By separating the discussion from the results, the authors were able to modify and improve.

*2. Add the limitation of your study in the discussion.*

**Response:** The authors included a discussion section, and the limitations of your study.

**DISCUSSION**

We used Krishnamoorthy and Wang's (2016) approach for establishing confidence intervals for the mean of a gamma distribution by using FQs in the case of the same distribution with excess zeros. Furthermore, we extended Yosboonruang et al.'s (2019) approach for building confidence intervals for distributions containing some zero observations by using Bayesian methods based on Jeffreys' rule and uniform priors. The results show that the Bayesian HPD interval with Jeffreys'rule prior performed well in terms of coverage probability and had the shortest expected lengths for estimating the mean and the difference between the means of delta-gamma distributions with equal sample sizes. For unequal sample sizes, the results of simulation for the difference between the means of two delta-gamma distributions for $n>m$ were similar to $n<m$. The proposed strategy can be utilized to help mitigate droughts or floods caused by insufficient or excessive rainfall, respectively. Similarly, the government could use our approach to control the output from dams when there is insufficient or too much rain. However, a limitation of the study is that we cannot apply our method to real-world data that does not contain zero observations, such as often occurs in rainfall data observations during the rainy season.

Best Regards,
Theerapong Kaewprasert, Sa-Aat Niwitpong, and Suparat Niwitpong
The authors